# Phase Transformation of Doped LiCoPO_4_ during Galvanostatic Cycling

**DOI:** 10.3390/ma13173810

**Published:** 2020-08-28

**Authors:** Wen Zhu, Dongqiang Liu, Catherine Gagnon, Vincent Gariépy, Michel L. Trudeau, Ashok Vijh, Karim Zaghib

**Affiliations:** 1Center of Excellence in Transportation Electrification and Energy Storage Hydro Québec, 1800 Boulevard Lionel-Boulet, Varennes, QC J3X1S1, Canada; liu.dongqiang@hydroquebec.com (D.L.); Gagnon.Catherine@ireq.ca (C.G.); gariepy.vincent@ireq.ca (V.G.); vijh.ashok@ireq.ca (A.V.); 2Department of Mining and Materials Engineering, McGill University, 845 Sherbrooke Street West, Montréal, QC H3A0G4, Canada; karim.zaghib@mcgill.ca

**Keywords:** doped-LiCoPO_4_, in situ XRD, phase transformation

## Abstract

In situ X-ray diffraction was employed to investigate the crystal structure changes in Cr/Si co-doped Li(Co,Fe)PO_4_ cathode material during a galvanostatic charge/discharge process at a slow rate of C/30. The evolution of the X-ray patterns revealed that the phase transformation between the Cr/Si-Li(Co,Fe)PO_4_ and Cr/Si-(Co,Fe)PO_4_ is a two-step process, which involves the formation of an intermediate compound of Cr/Si-Li_0.62_(Co,Fe)PO_4_ upon the extraction of Li ions from the pristine phase. Different from the previously reported two biphasic transition steps, the phase transformation of the Cr/Si-Li(Co,Fe)PO_4_ followed a solid solution and a biphasic reaction pathway at different stages of the delithiation/lithiation process, respectively.

## 1. Introduction

LiFePO_4_ (LFP) is a cathode material characterized by structural and thermal stability, low cost, and high safety, as well as a low operating voltage (3.5 V) and low conductivity [1,2,3]. To overcome the shortcomings deriving from the low operating voltage of LFP, the application of other members of the olivine family (LMPs), containing the transition metal M=Mn, Co, Ni, or a combination of them, has been explored [4,5,6,7]. Among them, LiCoPO_4_ (LCP) is of great interest, owing to its large theoretical specific capacity (167 mAh g^−1^) and high operating voltage 4.8 V (vs. Li/Li^+^). These properties result in a specific energy of ∼800 Wh kg^−1^ (i.e., ~25% higher than that of conventional cathodes in Li-ion batteries [5]).

LMPs and their charging end products, MPO_4_ (MP), have an olivine structure, with an orthorhombic unit cell and a Pnma space group (SG). Li ion intercalation/deintercalation processes occur in a one-dimensional channel along the [010] direction [8,9,10]. A two-phase reaction mechanism was first proposed for LFP, owing to the narrow single-phase ranges near the stoichiometry compositions of LFP and FePO_4_ (FP). Detailed studies have demonstrated that the LFP-to-FP reaction mechanism depends not only on the material’s intrinsic properties, but also on the size and orientation of the particles [11,12,13], the cycling rate [14,15], temperature [16], and the strains [17,18].

While the working mechanisms of LFP are well studied, it is likely to assume that the operation mechanisms of the LMP family members are the same. But previous studies on Li(Mn,Fe)PO_4_ and LCP have demonstrated different situations in these two cases. Perea et al. [19] examined the phase transformation process in the LiMn_y_Fe_1−y_PO_4_ system in relation to the Mn contents. They proposed a three-step phase change process for LiMn_y_Fe_1−y_PO_4_ (0.25 ≤ y < 0.75), which, in fact, showed five regions in the phase diagram, including two narrow solid solution (ss) regions near the two end products, plus three main regions. The phase regions are in the order of: (1) ss-Li_1−x_Mn_y_Fe_1−y_PO_4_; (2) biphasic domain related to Fe^3+^/Fe^2+^; (3) ss-intermediate phase; (4) biphasic domain of Mn^3+^/Mn^2+^; (5) ss-Mn_y_Fe_1−y_PO_4_ as the Li^+^ content decreased from 1 to 0. In addition, the phase transitions of Mn- and Fe-rich compositions (i.e., LiMn_y_Fe_1−y_PO_4_ with 0.75 ≤ y < 1 and 0 < y < 0.25) follow a pathway that pass through three regions in the phase diagram, i.e,—a solid solution, a biphasic, and another solid solution regions. The main difference between the mechanisms of Mn- and Fe-rich compositions lies in the biphasic region, which is attributed to the Mn^3+^/Mn^2+^ redox couple in the former and to the Fe^3+^/Fe^2+^ in the latter case. On the other hand, different phase evolution mechanisms have been proposed for LCP. Initially, a two-phase mechanism was suggested based on the ex situ X-ray diffraction (XRD) results—i.e., the phase transformation follows the reaction: LiCoPO_4_ ↔ CoPO_4_ + Li^+^ + *e*^−^ [4]. Later studies [7,20,21,22] introduced a stepwise phase separation mechanism by monitoring the phase evolution during cycling via the in situ synchrotron and in situ neutron diffraction techniques. This mechanism involved an intermediate phase of Li_0.6–0.7_CoPO_4_, and the transformation went through two biphasic regions: (1) LiCoPO_4_ ↔ Li_0.6_CoPO_4_ + 0.4Li^+^ + 0.4 *e*^−^; (2) Li_0.6_CoPO_4_ ↔ CoPO_4_ + 0.6Li^+^ + 0.6*e*^−^ (assume the intermediate phase is Li_0.6_CoPO_4_). In addition, an intermediate phase with a relatively low Li content (Li_0.2–0.45_CoPO_4_) was derived from the electrochemical experimental results [23]. In the present work, we investigated the working mechanism of a high-capacity Cr/Si co-doped LiCo_0.82_Fe_0.1_PO_4_ cathode via in situ XRD. The results confirmed the appearance of an intermediate phase, Cr/Si-Li_0.62_(Co,Fe)PO_4_, and revealed the concurrence of the solid solution and two-phase pathways during the charge/discharge process, for the first time.

## 2. Experimental

The electrochemical cell consisted of a Cr/Si co-doped LiCo_0.82_Fe_0.1_PO_4_ (simply called ‘LCFP’ hereafter)cathode, a Li anode, and a Celgard polypropylene separator, which were dried at 120 °C overnight under a vacuum. The electrolyte was composed of a 1.2 M LiPF_6_ solution in a 3:7 (wt.%) mixture of ethylene carbonate and ethyl methyl carbonate electrolyte, and 2 wt.% tris (trimethylsilyl) phosphite. The electrochemical measurements were carried out using a Biologic SP300 potentiostat (BioLogic Science Instrument, Seyssinet-Pariset, France), controlled by EC-Lab (V10.19, BioLogic Science Instrument, Seyssinet-Pariset, France). The experimental coin cell was cycled between 3.6 and 5 V at a rate of C/30 for three cycles. The first cycle was considered as a formation cycle, during which the cathode material was activated and the solid electrolyte interface (SEI) layer was formed, while the following cycles were considered more relevant to the electrochemical performance of the battery; therefore, the XRD patterns were only collected after the first cycle.

Cr/Si co-doped LiCo_0.82_Fe_0.1_PO_4_ was prepared via a high-energy ball milling of the raw materials, followed by a solid-state reaction. A slurry with 85 wt.% of LCFP active material was tape casted on a 15 µm thick Al foil. Detailed information about the preparation of the powder and cathode can be found in one of our previous works [24]. The in situ XRD patterns were collected from a standard 2032-coin cell with an 8 mm hole drilled on the top case. The Al current collector from the cathode film also acted as an X-ray window, and a silver paste was used to seal the Al window and the top part of the stainless-steel cell case. Figure 1 is a schematic diagram of the in situ cell. The in situ X-ray measurements were performed with a Smartlab diffractometer (Rigaku, Tokyo, Japan) using Co-Kα radiation at a step size of 0.04° and scan rate of 2.219°/min—i.e., 30 min/scan. The sample displacements were corrected by taking the Al diffraction peak as reference.

## 3. Results and Discussion

The XRD pattern of the as-made electrode was indexed to two phases: a major phase of LCFP (PDF: 04-007-4779) and a minor phase of Li_9_Cr_3_(PO_4_)_2_(P_2_O_7_)_3_ (PDF: 04-014-4369). No carbon peaks were observed, possibly owing to itsnanosize and overlapping with the LCFP peaks. The Al peaks were derived from the current collector. The XRD pattern was refined (using the Rietveld method) with an orthorhombic unit cell and a Pnma SG using the TOPAS software (Bruker, Billerica, MA, USA). The Fe, Cr, and Si dopants were assumed to occupy the Co sites with total occupancy fixed at 1. The peak shape, scale factor, lattice parameters, Lorentzian size and strain, as well as atomic positions and isotropic thermal parameters (B_eq_) were refined, see Figure 2. The refined lattice parameters were *a* = 10.2003(6), *b* = 5.9287(2), and *c* = 4.6971(1) Å. The crystal structure parameters are listed in Table 1. The refined LCFP was 98(1) wt.%, and the balance was Li_9_Cr_3_(PO_4_)_2_(P_2_O_7_)_3_, which was proven to be electrochemically inactive in our post-mortem analysis.

Figure 3 shows the voltage variation as a function of the Li concentration in a Li/LCFP half-cell cycled at a slow C/30 rate between 3.6 and 5 V. The blue curve (cycle A) represents the first cycle (after one formation cycle), for which the charge curve was considered to start at a Li concentration (*x*) of 1. As part of the applied current was consumed during the formation of the SEI and the related side reactions, rather than the removal of the Li ions, the *x*, calculated electrochemically, was smaller than the real value at the end of the charge, and consequently, the smaller *x* value at the end of discharge. As a result, the starting lithium content of the following cycle (cycle B) inherited the value at the end of the previous discharge, was smaller than the real value, and eventually led to the *x* < 0 at the end of charge, which is meaningless. Hence, the starting value of *x* was shifted to 1 for cycle B as the cell was fully discharged (Figure 3). The voltage curve in Figure 3 is asymmetric and with two distinct plateaus at ∼4.75 and ∼4.85 V during the charge; the plateaus, however, were much less distinctive during the discharge. The appearance of the two plateaus in the voltage profile was attributed to the two two-phase regions during the charge/discharge [7], but may also be related to the carbon content [23]. The charge and discharge capacities derived from the electrochemical data of cycle A were 168 and 121 mAh g^−1^, respectively. The charge capacity was slightly higher than the theoretical value of 167 mAh g^−1^, confirming that some side reactions occurred, such as the formation of SEI and the decomposition of the electrolyte at high voltage. During cycle B, the charge and discharge capacities were 147 and 120 mAh g^−1^, respectively. The decrease in charge capacity suggested a decrease in the side reactions as the cycling proceeded.

The small “blips” in the voltage–time plot in the discharge could have arisen from gaseous bubbles emanating during the slight decomposition of the SEI. This phenomenon is easily conceivable for carbonate electrolytes, in which the formation/decomposition of SEI can lead to the production of either CO_2_ or O_2_ from the carbonates/oxides in the SEI (typically having a complex composition). It is believed that these blips do not indicate a “bad” contact, because these usually show random and more persistent oscillations.

Figure 4 displays the evolution of the XRD patterns during two cycles and the corresponding voltage–time curve. In Figure 4a, all the unlabeled diffraction peaks belong to either LCFP or Cr/Si-(Co,Fe)PO_4_ (called CFP hereafter), or the intermediate phase. The peaks labelled with “*” were attributed to the stainless steel casing, the silver paste, and the Al window. The peak bars of LCFP and CFP are plotted at the bottom of the graph; the initial phase (LCFP) corresponds to the orange peak bars. Some of the LCFP peaks were noticeably upshifted during the charge and downshifted during the discharge. This indicates the contraction and expansion of the LCFP unit cell volume, which was caused by the extraction and insertion of the Li ions. Figure 4b shows the contour plot of the (020), (311), and (121) lines of the initial, the intermediate, and the final phases. In cycle A, at the end of the charge, the strongest line of LCFP, (311) was clearly seen in the plot, which indicated that a certain amount of LCFP was left in the electrode. The intensity of LCFP (311) was greatly reduced at the end of the charge in cycle B, suggesting a more complete phase transition. Thus, we decided to analyze the phase transformation of cycle B in detail. At the beginning of cycle B, the (020), (311), and (121) lines of LCFP were located at 2θ~ 35.3°, 42.1°, and 43.2°, respectively. As the charge proceeded, the three peaks shifted continuously toward high angles until reaching 4.831 V (Li^+^ = 0.657). Between 4.840 (Li^+^ = 0.624) and 4.851 V (Li^+^ = 0.224), the positions of the three peaks remained almost unchanged (they were located at 2θ∼ 35.7°, 42.5°, and 43.6°), suggesting the presence of a new phase and the depletion of LCFP. This new phase was formed through the delithiation of LCFP and assigned to Cr/Si-Li_0.62_(Co,Fe)PO_4_ (called L_0.6_CFP hereafter; PDF: 04-014-7340), which inherited the crystal structure of LCFP but possessed different lattice parameters. After the formation of the L_0.6_CFP phase, two peaks appeared at ~36.1° and 43.9°; they were ascribed to the (020) and (121) lines of the Cr/Si-(Co,Fe)PO_4_ phase. In addition, the (311) peak of the CFP phase was located at ~43.6°, overlapping with the (121) peak of L_0.6_CFP. The further charging of the cell resulted in a continuous increase in CFP peaks accompanied by the decrease in the strongest peak of L_0.6_CFP ((311)/42.5°), which eventually vanished at 4.862 V (Li^+^ = 0.190). This suggests that all of the L_0.6_CFP was transformed into CFP at Li^+^ = 0.190. This phase also has an orthorhombic unit cell with a Pnma SG. After the initial formation of the CFP, the phase grew monotonically until the end of the charge at 5 V (Li^+^ = 0.116), with its peak positions nearly unmoved. The phase change process occurring during the discharge was reversible. The CFP phase disappeared first, followed by L_0.6_CFP. Eventually, the diffraction peaks of the single-phase LCFP shifted continuously toward low 2θ, until returning to their original positions, which suggested the end of the discharge. The development of the in situ XRD patterns indicates that three phases are involved in the charge/discharge and the process is described with three regions in phase diagram, as indicated in Figure 5d. Region I contains the ss-LCFP and corresponds to a continuous shift in the LCFP peaks (Figure 4a,b), which is an indication of a solid solution mechanism between LCFP and L_0.6_CFP phase transition. L_0.6_CFP and CFP coexisted in region II: the peak positions of the two phases remained almost unchanged, whereas the quantities of the two phases varied in opposite ways (Figure 5c). Finally, region III contains the single-phase CFP.

The intermediate phase (with Li^+^ ~ 0.62) was similar to the Li_0.6-0.7_CoPO_4_ phase determined by in situ XRD [7,21], but considerably different from the Li_0.45–0.2_CoPO_4_ phase derived from the electrochemical measurement [23]. Despite the intermediate phase composition being consistent with those obtained from diffraction techniques reported in the literature, the phase change mechanisms did not coincide. Several researchers have reported the appearance of two biphasic regions during the charge/discharge process, indicating that the stepwise phase transformations follow a two-phase reaction route at every step. Our in situ XRD analyses revealed the existence of three regions in the phase diagram during cycling (Figure 5d). The evolution of the XRD patterns indicated that the phase transition from LCFP to L_0.6_CFP followed the solid solution pathway, and that the transformation from L_0.6_CFP to CFP underwent a two-phase reaction route. In fact, the phase transformation mechanism does not only depend on the intrinsic properties of the olivine structured materials, but also on other factors (e.g., the particle size, temperature, cycling rate, and coherent stain). Studies on the LFP have demonstrated that the variation in these extrinsic factors may modify or change the phase transition thermodynamics by reducing the miscibility gap in the LFP-FP system. For example, Li et al. [12] investigated the influence of particle orientation on the phase transition and miscibility gap in L_1−x_FP during nonequilibrium battery cycling. Their results showed that the phase transformation routes were not strictly single- or two-phase reactions but were orientation-dependent, even at low currents. They examined samples with dimensions in the [100] direction varying from 46 to 12 nm, and concluded that decreasing the dimension in the [100] direction can remarkably improve the solid solubility of both end solid solutions (Li_α_FePO_4_ and Li_1−β_FePO_4_) and decrease the lithium miscibility gap of LFP. The effects of size on solubility and on the miscibility gap have been previously discussed in the literature [25,26]. In a nano-size regime, the solubility limits of Li in two end phases (L_1−x_FP and L_y_FP) increased, while the miscibility gap in Li_1−x_FePO_4_ contracted systematically as the size of particles decreased. Liu et al. [15] focused on the influence of the cycling rate on the phase transformation of micro-sized particles. They revealed the development of a continuous solid solution, which extended from the two end-member phases into the thermodynamic miscibility gap. Consequently, the working mechanism of LFP under exceptional high cycling rates was considered to follow a facile nonequilibrium single-phase transformation pathway instead of the commonly recognized two-phase path. Meanwhile, Cogswell et al. [17] studied the effects of the coherency strain on the solubility and galvanostatic discharge of LFP nanoparticles through a reaction-limited phase-field model. Their calculations indicated that the coherency strain can strongly suppress phase separation during the discharge process. Islam et al. [8] explored the doping and defects in the LFP through atomistic modeling, showing that aliovalent dopants are not favored energetically. With the replacement of Fe^2+^ ions by various trivalent and divalent cations, they found that the energies of LFP with trivalent dopants are several electron volts higher than those with divalent dopants. In the case of our (Cr/Si)-LCFP, its crystal structure is shown in Figure 6, together with that of LCP. It is evident that the LCP structure is uniform; the octahedra are all CoO_6_, and there are no vacancies in the transitional metal layer. In contrast to LCP, the structure of (Cr/Si)-LCFP is not uniform; most of the octahedra are CoO_6_ or FeO_6_, whereas the rest are CrO_6_ or SiO_6_. Since the oxidation state of Fe^2+^ and Co^2+^ is the same, and the ionic radii difference is small, the substitution of Fe^2+^ to Co^2+^ induced very small local distortion, and no vacancy was involved. On the other hand, the replacement of Co^2+^ by the smaller trivalent/tetravalent Cr^3+^/Si^4+^ led to the formation of smaller CrO_6_/SiO_6_ octahedra as well as vacancies on transition metal sites, and all these nonuniform areas resulted in greater local distortion and strain energy. This strain energy could have played a similar role to the coherent strain during the phase transformation of LCFP into L_0.6_CFP (i.e., the suppressed phase separation) and changed the phase transition pathway from two-phase reaction to one-phase. In the case of the transformation of L_0.6_CFP into CFP, it is possible that the strain energy induced by the dopants was not enough to suppress the phase separation; thus, the phase transformation would have still followed the nucleation and growth route.

The in situ XRD dataset is also refined using the Rietveld method with crystal structure parameters (atomic positions, thermal parameters) fixed, and the R_wp_ of the refinements ranged from 6.1 to 9.0. Figure 5a,b display the lattice parameter variations vs. the Li content during cycle B. The differences between the *b* and *c* values of the three phases were small, whereas the differences between the *a* values of the CFP and the other two phases were large. Table 2 lists the lattice parameters of the pristine LCFP phase, the average values of L_0.6_CFP and CFP, and differences between them. The lattice parameters of L_0.6_CFP and CFP remained nearly constant during the charge and discharge, owing to their approximately constant compositions, whereas the lattice parameters of LCFP changed continuously during cycling because of its solid solution nature. It is clear that the lattice parameter changes are anisotropic, and so are the stresses induced. The differences in volume were mainly caused by the changes in the *a*-axis. The volumes contracted by 2.57% from LCFP to L_0.6_CFP, and by 4.39% from L_0.6_CFP to CFP, a total of ~6.85% from LCFP to CFP (similar to the volume change from LFP to FP (6.81%)) [27]. This relatively large volume contraction/expansion is likely to crack the cathode material during cycling, especially in the all solid-state battery, thus deteriorating the material and leading to its eventual failure.

## 4. Conclusions

In situ XRD was used to monitor the structural changes in Cr/Si-LiCo_0.82_Fe_0.1_PO_4_ during the charge–discharge process. Its phase evolution revealed the presence of single-phase and two-phase reaction stages upon the lithiation/delithiation process of this Cr/Si-LCFP. The characteristics of the intermediate phase of L_0.62_CFP were in good agreement with published data; however, the combined solid solution and biphasic reaction mechanism is different from the previously reported two biphasic reaction mechanism. The difference may be caused by the substitution of Co/Fe by aliovalent ions of Cr^3+^ and Si^4+^, which could have introduced strain in the structure and, hence, partially changed the phase transformation route during cycling.

## Figures and Tables

**Figure 1 materials-13-03810-f001:**
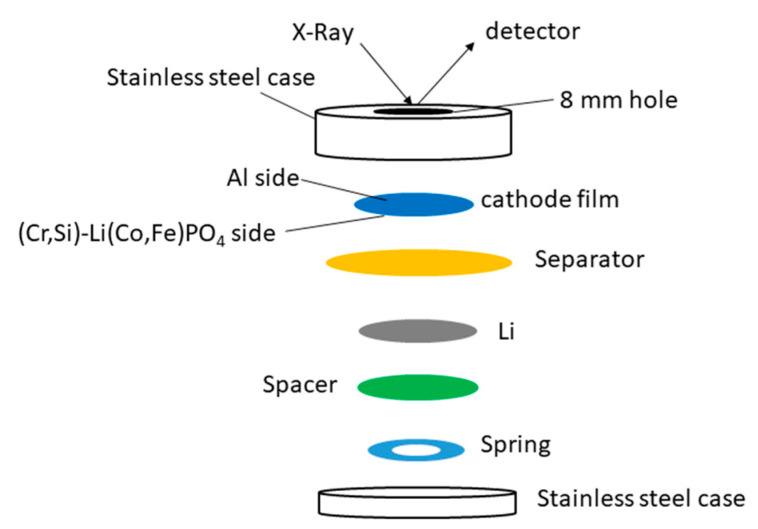
A schematic diagram of in situ XRD cell.

**Figure 2 materials-13-03810-f002:**
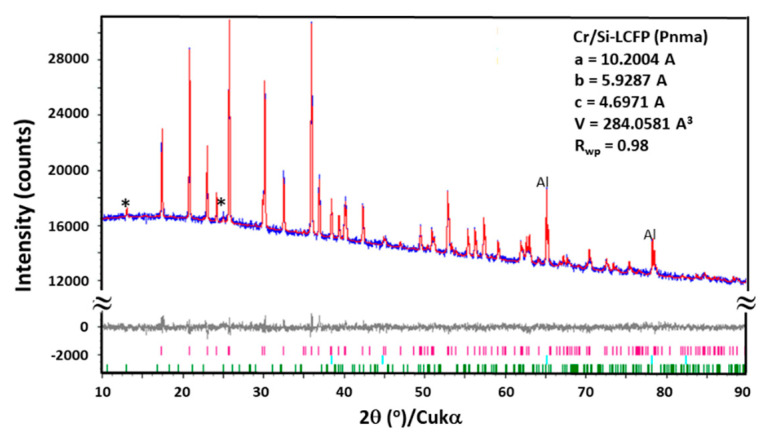
Measured and refined XRD patterns of the as-made Cr/Si-LCFP electrode. Blue = observed; red = calculated; grey = difference; magenta = LCFP; turquoise = Al; green/* = Li_9_Cr_3_(PO_4_)_2_(P_2_O_7_)_3_.

**Figure 3 materials-13-03810-f003:**
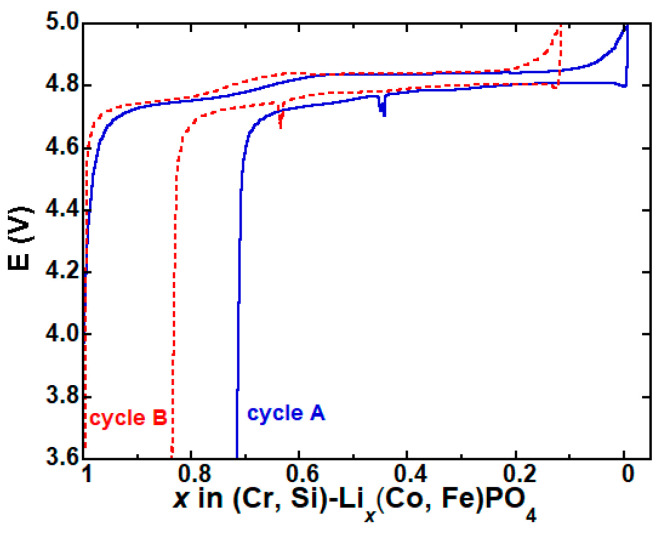
Variation in cell voltage vs. Li concentration during cycling.

**Figure 4 materials-13-03810-f004:**
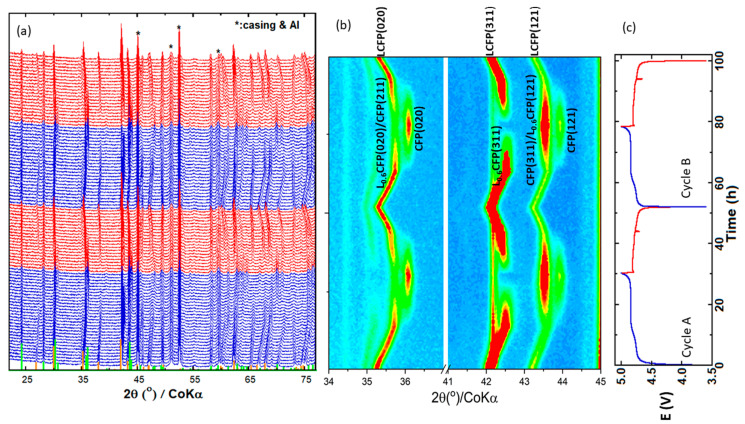
Evolution of the XRD patterns of a Li/LCFP half-cell cycled at a rate of C/30. (**a**) Full 2θ range; (**b**) contour plot of the (020), (311), and (121) lines; (**c**) corresponding voltage–time profile (the displayed cycles correspond to the 2nd and 3rd cycles of the cell, while the 1st formation cycle is not plotted). Blue = charge; red = discharge. Peak bars at the bottom of (**a**): orange = LiCoPO_4_/PDF-04-007-4779; green = CoPO_4_/PDF-04-014-7341.

**Figure 5 materials-13-03810-f005:**
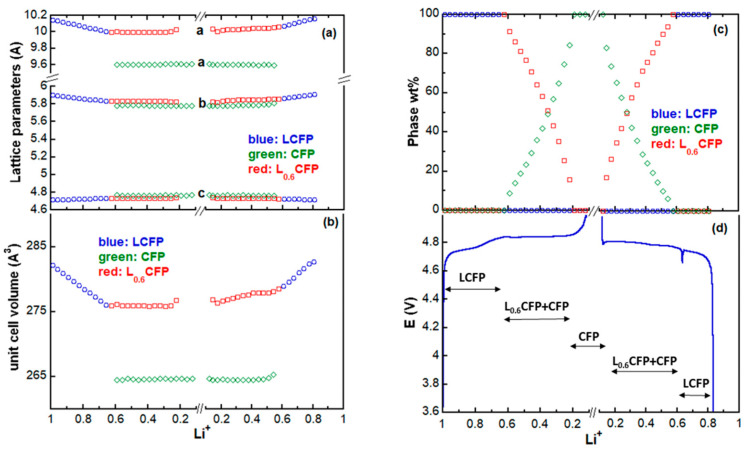
Variation of lattice parameters in cycle B: (**a**) lattice parameters; (**b**) unit cell volumes; (**c**) phase wt.% and (**d**) cell potential vs. the Li content during the charge/discharge process.

**Figure 6 materials-13-03810-f006:**
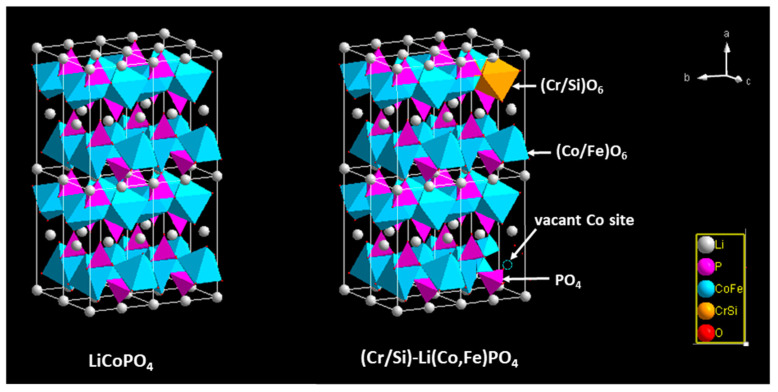
Schematic crystal structure diagrams of LiCoPO_4_ and (Cr/Si)-Li(Co,Fe)PO_4_.

**Table 1 materials-13-03810-t001:** Refined crystal structure parameters.

Atoms	Wyckoff Sites	x	y	z	Occupancy	B_eq_
Li	4a	1/2	1/2	1/2	1	1.883(0)
P	4c	0.0925(8)	1/4	0.4133(7)	1	0.682(9)
Co/Fe/Cr/Si	4c	0.2785(4)	1/4	0.9762(3)	1	1.142(0)
O	4c	0.0953(9)	1/4	0.7473(2)	1	1.226(0)
O	4c	0.4532(0)	1/4	0.2176(5)	1	1.226(0)
O	8d	0.1640(9)	0.0413(8)	0.277491)	1	1.226(0)

R_wp_ = 0.98; GOF (goodness of fit) = 1.18.

**Table 2 materials-13-03810-t002:** Average lattice parameters and their differences between the three phases.

	*a (A)*	*b (A)*	*c (A)*	*V (A^3^)*	Δ*a*%	Δ*b*%	Δ*c*%	Δ*V*%
LCFP (Li = 1)	10.2003(6)	5.9287(2)	4.6971(1)	284.058(1)				
L_0.62_CFP_ave_	10.0191(5)	5.8387(0)	4.7306(3)	276.767(3)				
CFP_ave_	9.6004(0)	5.7852(2)	4.7642(4)	264.608(1)				
LCFP–L_0.62_CFP_ave_					−1.77(7)	−1.51(8)	0.71(4)	−2.56(7)
L_0.62_CFP_ave_–CFP_ave_					−4.17(9)	−0.91(6)	0.71(0)	−4.39(3)
LCFP–CFP_ave_					−5.88(2)	−2.42(0)	1.42(9)	−6.84(7)

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
