# Peer review of "Phase Transformation of Doped LiCoPO4 during Galvanostatic Cycling"

_materials, 2020, doi:10.3390/ma13173810_

Round 1

Reviewer 2 Report

The paper reports the phase transformation of Cr/Si doped LiCo0.82Fe0.1PO4 (LCFP) during cycling at a low rate. The in situ X-ray analysis was conducted to reveal 3 phases involved in the cycling with 3 regions in the phase diagram. The solid solution phase transformation from LCFP to Li0.6CFP is followed by the co-existence of Li0.6CFP and CFP and final transformation from Li0.6CFP to CFP via a two-phase reaction route during charging, which reversely occurs during discharging. Difference from the reported literature, the authors did not observe two biphasic regions, which they attribute to the introduction of strain to the structures via substitution of Co/Fe by aliovalent ions of Cr3+ and Si4+. The finding is solid and the results contribute to the understanding of the structure change during cycling of the cathode for LIBs. I think the paper is interesting for the readers in the field. However, minor revision is required for the present form of the draft and there are some points to be addressed before acceptance for the publication.

1) In the introduction, it is mentioned that the stepwise phase separation mechanism for LCP with an intermediate phase of Li0.6-0.7CoPO4 and transformation through two biphasic regions. The paper is, in contrast, reports a biphasic region in the phase diagram in the cycling. Hence, more detailed information of the two biphasic regions is recommended to add to the introduction for better comparison with the finding shown later in the paper.

2) The use of in situ XRD analysis is a key point for obtaining the structure change during charge/discharge, hence, an addition of an illustration of the structure or picture of the coin cell for the in situ XRD analysis is recommended to the experimental section.

3) A model (illustration) of the unit cell of LiCo0.82Fe0.1PO4 and after Cr/Si doping should be added for the benefit of the reader and direct view of the structure.

4) The paper discussed well and in detail the phase transformation with in situ XRD results associated with LCFP, Li0.6CFP, and CFP phase (Figures 3, 4). The XRD refinement (Figure 1) indicates the presence of Li0.9Cr3(PO4)2(P2O7)3 as a minor phase (2 wt%) aside from the main LCFP phase. Is there any contribution or involvement of Li0.9Cr3(PO4)2(P2O7)3 to the phase transformation during cycling? Can the authors comment on this matter?

5) The strain caused by doping of aliovalent ions of Cr3+ and Si4+ is referred to as a possible reason for observing the one biphasic region in the phase diagram followed the solid solution mechanism to convert LCFP to Li0.6CFP. If it is possible, I recommend extracting the strain caused by Cr3+/Si4+ doping from the XRD analysis, and its impact on the phase transformation can be discussed more quantitatively.
